# Viroid-infected Tomato and Capsicum Seed Shipments to Australia

**DOI:** 10.3390/v11020098

**Published:** 2019-01-24

**Authors:** Fiona Constable, Grant Chambers, Lindsay Penrose, Andrew Daly, Joanne Mackie, Kevin Davis, Brendan Rodoni, Mark Gibbs

**Affiliations:** 1Agriculture Victoria Research, Department of Jobs, Precincts and Regions, AgriBio, 5 Ring Road, Bundoora, VIC 3083, Australia; brendan.rodoni@ecodev.vic.gov.au; 2NSW Department of Primary Industries, Elizabeth Macarthur Agricultural Institute (EMAI), Woodbridge Road, Menangle, NSW 2568, Australia; grant.chambers@dpi.nsw.gov.au (G.C.); andrew.daly@dpi.nsw.gov.au (A.D.); 3Australian Government Department of Agriculture and Water Resources, 7 London Circuit, Canberra City, ACT 2601, Australia; lindsay.penrose@agriculture.gov.au (L.P.); joanne.mackie@agriculture.gov.au (J.M.); kevin.davis@agriculture.gov.au (K.D.)

**Keywords:** pospiviroid, viroid, PSTVd, tomato, capsicum, seed, trade, detection

## Abstract

Pospiviroid species are transmitted through capsicum and tomato seeds. Trade in these seeds represents a route for the viroids to invade new regions, but the magnitude of this hazard has not been adequately investigated. Since 2012, tomato seed lots sent to Australia have been tested for pospiviroids before they are released from border quarantine, and capsicum seed lots have been similarly tested in quarantine since 2013. Altogether, more than 2000 seed lots have been tested. Pospiviroids were detected in more than 10% of the seed lots in the first years of mandatory testing, but the proportion of lots that were infected declined in subsequent years to less than 5%. Six pospiviroid species were detected: *Citrus exocortis viroid*, *Columnea latent viroid*, *Pepper chat fruit viroid*, *Potato spindle tuber viroid*, *Tomato chlorotic dwarf viroid* and *Tomato apical stunt viroid*. They were detected in seed lots exported from 18 countries from every production region. In many seed lots, the detectable fraction (prevalence) of infected seeds was estimated to be very small, as low as 6 × 10^−5^ (~1 in 16,000; CI 5 × 10^−6^ to 2.5 × 10^−4^) for some lots. These findings raise questions about seed production practices, and the study indicates the geographic distributions of these pathogens are uncertain, and there is a continuing threat of invasion.

## 1. Introduction

Trade in plant materials, especially seeds, tissue cultures and nursery stock, can spread pathogens, changing their epidemiology and geographic distribution. Pospiviroids are emerging pathogens with several species discovered in solanaceous crop plants in the past 20 years, and more species are spreading to countries and continents where they have not been previously recorded [1,2,3,4,5,6,7,8,9]. Pospiviroids have been detected in cultivated ornamental plants in nurseries, and it was proposed that ornamental plants are the source of the infections in crops [10,11]. In Australia, *Pepper chat fruit viroid* (PCFVd) has been detected once in a tomato (*Solanum lycopersicum*) crop, and *Potato spindle tuber viroid* (PSTVd) has been detected several times in chilli (*Capsicum* spp.) and tomato crops [12,13,14,15,16]. Rather than focusing solely on ornamental plants, Australian governments have considered infected tomato, capsicum and chilli seeds as the likely sources of the infections. Pospiviroids are transmitted through the seed of capsicum and tomato [8,17,18,19,20,21,22,23,24,25], and these vegetable crops in Australia are almost always grown from imported seed. Clear evidence links a few outbreaks in Australia and other countries to seed, and seed has been suspected to be the source of several other outbreaks in light of the surrounding circumstances [13,14,20,26,27,28,29,30].

Australia started testing small numbers of imported tomato seed lots for PSTVd in 2008. In 2012 and 2013, testing of all imported tomato and capsicum seed lots for several pospiviroid species became mandatory, and many more were tested. Seed lots were either tested in the exporting country before being sent to Australia or tested upon arrival in Australia, and those lots that were found to be infected were not permitted into Australia for planting. This paper presents results from the two Australian laboratories where imported capsicum and tomato lots were tested upon arrival. The testing reported here was done on a commercial basis, not as a research project, and the test results and related data are not entirely complete, as they were not always systematically collected. Regardless, the data is sufficient to show that traded seed lots were contaminated with pospiviroid-infected seeds (‘infected seed lots’) and the trade in infected seed lots was more significant than previously understood [31]. We know of only one other survey of a similar scale [32]. Our testing also showed changes in the incidence of infected seed lots that appeared to be related to Australian regulation. This study calls attention to the threat of invasion by pospiviroids and the connection between seed trade and the epidemiology of these seed-borne viroids. The study also underscores the need to measure and explain the fraction of infected seeds in seed lots, as this parameter has a bearing on epidemiology, detection and the phytosanitary preparation of seed lots.

## 2. Materials and Methods

Each ‘seed lot’ was identified by the importer as consisting of a single cultivar produced in one country. Imported capsicum (including pepper and chilli) and tomato lots sent to Australia were tested by the diagnostic laboratories of the New South Wales and Victorian state governments at the Elizabeth Macarthur Agricultural Institute (EMAI; New South Wales Department of Primary Industries, Menangle, Australia) and the Crop Health Services laboratory (CHS; Agriculture Victoria Research, Department of Jobs, Precincts and Regions, Bundoora, Australia). Testing was introduced in June 2008 on tomato seed only, but it was not mandatory until 2012, and the complete suite of tests was not implemented until early 2013. Mandatory capsicum seed testing started at the end of 2012. The CHS laboratory compiled test results between July 2008 and April 2016 and the EMAI laboratory compiled equivalent results between 2013 and 2016.

Every lot was sampled by drawing 20% of the total lot by weight, up to a maximum of about 20,000 seeds. If the primary sample was larger than 400 seeds it was divided into subsamples consisting of at most 400 seeds, also drawn by weight. Following this sampling process, if a lot comprised more than 100,000 seeds, then a primary sample of 20,000 seeds was taken, and was divided up into 50 sub-samples. If a lot consisted of 100 seeds, as was the case for the smallest tested imports, then a sample of just 20 seeds was taken, and this sample was not divided into subsamples.

Each subsample was tested independently. Subsamples were crushed to a powder in an extraction bag (Bioreba) using a hammer or in a 50 mL centrifuge tube using a 14 mm stainless steel ball bearing and a 2010 Geno/Grinder® (AXT, Sydney, Australia). Total nucleic acids (TNA) were extracted following the method of Hoshino et al [33] and conventional reverse-transcription PCR (RT-PCR) assays were done. Between 2008 and 2012, the extracts were tested using two PSTVd specific primer sets [13,34]. In 2012, the testing changed and the extracts were tested using specific primers for PSTVd [13] and *Columnea latent viroid* (CLVd) [35] and the universal primers (Pospi) for pospiviroids [3]. This range of tests is believed to be capable of detecting all the known pospiviroid species that infect solanaceous crops.

Seed handling, TNA extractions, RT-PCR-mix preparation, RT-PCR tests and gel-electrophoresis were done in separate work areas and using segregated work flow procedures to minimise the risk of contamination. Airflows were controlled, aerosol resistant pipette tips were used and equipment was not moved between work areas.

A positive control was used for every set of RT-PCRs that consisted of TNA extracted from infected plant tissue or from a previously identified pospiviroid-infected seed lot. In some instances, DNA from a plasmid containing a cDNA insert of the targeted viroid was used as a positive control. Negative water controls without TNA nucleic acids were also included in every set of RT-PCRs. After amplification, 10 µL of each RT-PCR was run on a 1.5% agarose gel, stained with SYBR® Safe (Thermofisher) or GelRed® (Biotium) nucleic acid stain and visualized with ultraviolet (UV) excitation. PCR products of the expected size were sequenced directly in both directions and primer sequences were removed from the consensus sequences as were poorly resolved regions.

Viroid sequences were identified, and their associations were investigated by searching the GenBank non-redundant database for matches using BLASTN and by examining dendrograms generated by the minimum evolution method implemented in the BLASTN suite of programs [36,37]. The identities and similarities of sequences from different seed lots were also investigated by aligning using the programs MAFFT, MUSCLE and CLUSTAL OMEGA [38,39,40,41].

The underlying fraction of infected seeds in a seed lot was estimated using a Markov chain Monte Carlo (MCMC) method employing sampling models that followed binomial distributions. The models took account of the numbers of: (i) Seeds in subsamples, (ii) subsamples that were tested from the lot, (iii) subsamples that tested positive for any pospiviroid species, and (iv) subsamples that tested negative. This approach took advantage of the results for many lots where some subsamples tested positive, and other subsamples taken from the same primary sample tested negative. As the subsamples from all but the smallest lots consisted of about 400 seeds, the detection of positive and negative subsamples from the same primary sample indicated that the primary sample contained only a small fraction of infected seeds. Credible intervals for estimates of the fraction of infected seeds were obtained directly from stationary MCMC samples. The analyses assumed each subsample consisted of 400 seeds or less, where this was known, and that pospiviroids were detected without error, that is, the rates of false negative and false positive test results were zero.

## 3. Results and Discussion

### 3.1. Incidence of Infected Seed Lots

Over four years, pospiviroids were on average detected in 6.5% (36/553) of capsicum seed lots tested by Australian laboratories, and over eight years, pospiviroids were on average detected in 5.7% (91/1562) of tomato lots tested. The proportions of all tested lots that were infected varied considerably from year to year (Figure 1 and Figure 2). Some of the variation, but not all of it, appeared to be a stochastic effect related to small sample sizes, especially during the period between 2008 and 2012, when only small numbers of tomato seed lots (28 lots) were tested (Figure 2). In 2012 testing of tomato seed imports became mandatory, after which many more tomato lots were tested (Figure 1), and estimates of the proportion of infected tomato lots became more accurate. In 2012 about 14% of tomato lots were found to be infected, which is the peak detection point for the period 2012 to 2016.

Capsicum seeds were not tested for pospiviroids until 2013, when testing became mandatory. The peak year for detections of infected capsicum lots was 2013, when 12% of capsicum lots were found to be infected, a figure which is close to that for tomato seeds.

It is significant in our view that the proportion of tomato and capsicum lots found to be infected declined to less than 5%, tending to less than 2%, after the peak years (Figure 2). We interpret these declines to be responses to the imposition and maintenance of mandatory testing, as the quarantine regulatory measure. Taken together, the declining trends in the proportion of infected lots, suggested that the health of the seed supplied to Australia was improving. It is not known why the health of the seed supply improved. Seed production practices may have improved, or alternatively seed companies may have chosen to send higher quality seed, and some seed lots may have been diverted to other markets. Two major laboratories outside of Australia changed the seed testing methods they were using after 2012, which may have led to more detections outside of Australia and might then help explain the improving trend.

Similar trends were seen when data from the EMAI laboratory was considered on its own, but when the testing results of the CHS laboratory were considered separately, the trend of reducing detections of infected capsicum seed was not visible. The two laboratories tested lots imported by a different range of companies, and the CHS laboratory tested fewer capsicum lots than the EMAI laboratory in most years. The proportion of infected capsicum lots detected by the CHS laboratory remained quite high over most of the years (2013–2016), being greatest in 2016 at 14%.

The choices of seed trading companies appeared to be a major factor affecting the variation in testing and detections. The numbers of lots imported by almost every seed trading company varied greatly from year to year, as estimated from the lots tested in Australia, and likewise the numbers exported from production countries varied greatly from year to year. There were some consistencies however, notably from 2008 to 2011, when very few seed lots were tested as seed companies and exporting countries chose to certify that almost all tomato seed lots sent to Australia were free of pospiviroids. This was done using a certification option that did not require testing.

Over the entire period (2008–2016), seed companies could choose to test in laboratories in countries other than Australia, and the two Australian laboratories did not perform all the testing in any year from 2012 onwards. It is thought that the Australian laboratories tested the majority of imported lots in 2012 and 2013, but after that time laboratories in other countries did the majority of testing, partly explaining the decline in the numbers tested (Figure 1). Laboratories in other countries detected infected seed lots, but the numbers tested and detected are not known.

We believe this report to be the first to describe large-scale regular testing of commercially traded capsicum seed for viroids, and the first to detail the pospiviroid threat from traded capsicum (chilli and pepper) seed lots. Pospiviroids have been detected before in commercially traded tomato seed lots [42,43], but the results of regular testing have not been published in sufficient detail to allow the scale of the problem to be recognised. Bruinsma et al. [32] tested many tomato seed lots and found an average incidence of 1.9% over four years, and in the last year of testing reported a detection rate of nearly 7%. The difference between the averages reported here and the results of Bruinsma et al. [32] is most likely explained by differences in sampling. The primary samples tested by Bruinsma et al. [32] consisted of only 3000 seeds, so the chances of detecting lots containing very few infected seeds were substantially lower. The different incidences reported by Bruinsma et al. [32] might also be explained by differences in the detection methods or in the sources of the seed. The authors did not publish the sources of the seed lots that were tested, and it is not clear if they were internationally traded seed lots.

### 3.2. Pospiviroid Species Identification and Trends

For most detections, sequences more than 150 nt long that were of sufficient quality to identify the viroid species were obtained. Some sequences were found to closely match ones in the GenBank database across most of their length, and many obtained from different seed lots were identical or nearly identical with each other (Appendix A). Although misincorporation during PCR could not be discounted as the origin of some variant nucleotide positions, the sequence comparisons suggested that the reverse-transcription and PCR reactions had introduced few errors.

Six pospiviroid species were detected in tomato seed lots, four of which were also found in capsicum seed lots. Two new associations with seed were discovered. PCFVd was first found in tomato seed in the course of the testing [44], and CLVd was first found in pepper seeds that were tested for export to Australia by the California Seed and Plant Lab (S. Pannu, Pers. Comm. 2013); this viroid species was subsequently also detected in capsicum seed by one of the Australian laboratories.

Before the testing began, it was anticipated that PSTVd would be detected in some tomato seed lots. Detection of the other pospiviroid species was unexpected, and probably occurred because the primer sets used were capable of amplifying cDNA from a wider set of pospiviroid species, and because larger primary samples were tested.

The numbers of detections of each pospiviroid species rose and fell over the years (Figure 3) suggesting changes in the supply of seeds or in infections of seed production crops. PSTVd was detected in 46 capsicum and tomato lots altogether and was detected most often in 2013, after which detections of the viroid declined (Figure 3). PCFVd was detected almost as frequently, in 43 lots altogether, and it was the most commonly detected viroid species from 2014 onwards. PCFVd was found most often in 2014 in both plant species, and also appeared to decline after that peak. *Citrus exocortis viroid* (CEVd) and CLVd were both detected in 19 lots, and both viroids were found most often in 2013. Detections of *Tomato apical stunt viroid* (TASVd) and *Tomato chlorotic dwarf viroid* (TCDVd) were relatively rare and neither was found in capsicum seed. TASVd and TCDVd detections peaked in 2012 and 2013 respectively, and these viroids were not detected after 2013. Mixed contaminations of two or more pospiviroid species were detected in 11 lots.

Database searches and alignments suggested that a diversity of variants of several viroid species had been detected. As an example, the partial sequences from 18 CLVd detections (322–369 nt) had 90–99% identity with each other and 85–99% identity with other CLVd sequences recorded in GenBank; sequence comparisons suggested that different CLVd isolates detected in seed would be placed in different clusters of the known CLVd isolates represented in the database (Appendix A).

Database searches and alignments suggested that a diversity of variants of several viroid species had been detected. As an example, the partial sequences from 18 CLVd detections (322–369 nt) had 90–99% identity with each other and 85–99% identity with other CLVd sequences recorded in GenBank; sequence comparisons suggested that different CLVd isolates detected in seed would be placed in different clusters of the known CLVd isolates represented in the database (Appendix A).

Similarly, the partial sequences from 42 PSTVd detections (155–359 nt) represented many variants that had between 87–100% identity with each other; comparisons indicated that some of the PSTVd isolates that had been detected should be placed in known clusters of PSTVd isolates, and that some probably fell outside the recognised clusters (Appendix A). PSTVd sequences from the seed detections were considered especially interesting because there had been outbreaks of the viroid in chilli and tomato crops in Australia in the past two decades [12,13,14,15,16]. Comparisons with sequences from eight PSTVd isolates previously reported in Australia showed that six of the seed detections were identical (100% sequence identity) to Australian isolates from the Naaldwijk/Chittering cluster, and a further three had 98–99% identity with sequences from that cluster (Appendix A [14,15]). The remaining 33 PSTVd sequences detected in seed did not closely match with PSTVd sequences previously reported in Australia.

The partial sequences from 23 PCFVd detections had between 90–100% identity with each other, and several PCFVd sequences sequences from isolates from capsicum and tomato seed lots were identical (Appendix A). The partial sequences from two TASVd detections (195 nt) were 99% identical to each other and had 89–94% identity with 30 sequences recorded in GenBank (Appendix A). The partial sequences from eight TCDVd detections (150 nt) had 95–100% identity with each other (Appendix A). Four identical TCDVd sequences originated from four seed companies and three different geographic regions.

### 3.3. Geographic Origins

Pospiviroid-infected seed lots were produced in every major seed production region and in more than 18 countries including three countries in Africa, one in North America, four in Central and South America, six in Asia, two in Europe and two in the Middle East. Most capsicum and tomato seed lots sent to Australia came from Europe and North America, and infected seed lots were exported from these two regions (Figure 4). Although far fewer seed lots were exported to Australia from Africa, Asia, South and Central America and the Middle East, our data indicate that many infected seed lots were exported from those regions, and suggest seed production crops were more commonly infected in those regions (Figure 4), especially in Africa and Asia.

The geographic distributions of the pospiviroid species indicated by our detections in seed were largely consistent with known distributions, but there were surprises. CLVd was known to occur in Europe, North and Central America and Africa [45,46], but our data from seed testing suggested that it also occurred in the Middle East.

Our data suggest there is uncertainty about the global distributions of pospiviroid species outside of Australia. Seed production crops must be infected to produce the infected seed lots that were detected, but we found no reports of these infections in the scientific literature and the relevant databases. It seems likely that infections of crops, including seed production crops, often go unnoticed or unreported.

### 3.4. Fraction of Infected Seeds within Seed Lots

When an infected seed lot was detected, testing almost always showed that only a small fraction of the seed was infected, that is, the infected lots appeared to be composed largely of uninfected seeds and a relatively small number of infected seeds. The fraction of infected seeds was estimated to range from 5 × 10^−3^ to 6 × 10^−5^ across different lots. Many lots were estimated to have infected fractions at the lower end of this range of best estimates (<5 × 10^−4^). The lowest of the best estimates, 6 × 10^−5^, which was the lowest possible best estimation given the sampling procedure, is equivalent to one infected seed in 16,000 healthy seeds.

Figure 5 shows credible intervals for a selection of estimates (blue bars). For seed lots with a best estimate of 6 × 10^−5^, the credible intervals (P2.5–P97.5) ranged from 2.5 × 10^−4^ to 5 × 10^−6^. At the opposite end of the scale, the greatest best estimate of 5 × 10^−3^ (1 infected seed in 200) had credible intervals (P2.5–P97.5) of 1.4 × 10^−2^ to 1.4 × 10^−4^. No trend was seen when estimates of the infected fraction (red points) were plotted against the size of the seed lot (*x*-axis; Figure 5).

Bakker et al. [47] detected PSTVd in two commercial tomato seed lots and estimated that the fraction of infected seeds in both lots was greater than 1 × 10^−3^. In contrast, we found that the fraction of infected seeds in most lots tended to be at least ten-fold lower, and that in some lots it was more than 100-fold lower (Figure 3). We believe our results may accord with earlier reports of low levels of infected seed in commercial seed lots that were difficult to detect [48].

Detecting infected seeds when they are present only as a very small fraction of a lot, as reported here, is only possible when large samples of several tens of thousands of seeds are tested. Samples of tens of thousands of seeds are recommended for other seed-borne and seed-transmitted pathogens [49,50,51], indicating that other pathogens are also detected in very small fractions of lots. However, the International Seed Federation currently recommends testing samples of only 3000 tomato seeds for pospiviroids [52]. This recommendation should be reviewed.

### 3.5. Epidemiological Implications

Pospiviroid infections have been reported in capsicum and tomato crops internationally, and before Australia imposed mandatory testing there were several incursions of PSTVd in Australian crops, as well as one PCFVd incursion [1,2,3,4,5,6,7,8,9,12,13,14,15,16,17,26]. Given the numbers of infected seed lots detected in tests reported here, it is reasonable to infer that crop infections in Australia and elsewhere were caused by infected seed.

In Australia, more than 250 million tomato and capsicum seeds on average are imported for planting each year, and we presume other countries are importing similar or larger quantities. In view of these traded volumes and the numbers of infected seed lots (Figure 1 and Figure 2, [32,42,43]), it is likely infected seeds are planted in countries where these vegetables are grown. Pospiviroids may be infecting crops undetected or unreported, and until sufficient surveillance is done, the distributions and incidences of these plant pathogens outside Australia are uncertain, and may be under-estimated.

The factors connecting the epidemiology of the viroids and the production of infected seed lots need to be investigated. One factor is the production and planting of seed lots with very small numbers of infected seeds. These seed lots may escape detection but still spread the viroids.

We envisage two scenarios that might produce such seed lots. In the first scenario a few plants are infected in a seed production crop, and these plants produce few seeds, so when seeds are extracted, small numbers of infected seeds are included in a lot, but most of the seeds in the lot are uninfected. An alternative less obvious scenario is that a healthy uninfected seed lot is inadvertently contaminated with infected seeds from an infected lot.

The first scenario would occur if a viroid is present on a farm but does not spread well, or if infected plants are removed as a disease prevention measure, but this measure is not fully effective and some infected plants are missed. Infected plants are sometimes symptomless or only mildly affected [15,26], so it is likely many would go unrecognised in crops. Additionally, pospiviroid-infected plants have been reported to produce small fruit and few seeds [48,53]. Currently no in-depth analysis of these factors, including the incidence of pospiviroids across many crops, has been published.

The second scenario of inadvertent cross-contamination seems equally plausible, as large quantities of seed are processed and we could find nothing published to show that contamination of one lot with some seed from another lot is normally prevented during extraction and handling. The standards written for the Good Seed and Plant Practices (GSPP) system [54] indicate that separation of batches of tomato fruit and seed during processing is required under the system for the control of the seed-borne pathogen *Clavibacter michiganesis* subsp. *michiganensis*. However, most internationally traded tomato seed is not produced under the GSPP system. The specified requirements for separation in the GSPP system suggest that the industry recognises the potential for inadvertent contamination. Normal industry practices that might lead to cross-contamination need to be investigated to determine the significance of the problem. Inadvertent cross-contamination producing lots with very small numbers of infected seeds, which appears to be a likely explanation of the tiny fractions we detected, may also affect production of seeds of other species.

## 4. Conclusions

Others have argued that commercially produced vegetable seeds are unlikely to be responsible for most outbreaks of pospiviroids [10,52]. We believe that the evidence [1,2,3,4,5,6,7,8,9,12,13,14,15,16,17,18,19,20,21,22,23,24,25,26,27,28,29,30,31,32,42,43] indicates infected seed is responsible for long-distance transport of pospiviroids and for moving these pathogens to areas where previously they were not present, even though the rate of seed-transmission is probably low. Our study added another dimension to the weight of evidence by indicating a route of transport that is active, and indicating the potential for invasion is real.

Research is needed to identify and characterise the mechanisms and routes that allow viroids and other pathogens to move from one region to another, and one country to another. Research is also needed to discover which pests and pathogens are being carried by traded plant materials. Without understanding of these factors, pathogens will invade places where they were not previously present, crop yields will be affected and countries may not institute effective regulations. The Australian testing produced surprising results, showing that several pospiviroid species were circulating in commercially-produced internationally-traded capsicum and tomato seeds. Before large numbers of seed lots were subjected to mandatory testing, these hazards and their magnitude were largely unrecognised. In fact, the numbers of infected seed lots have been so great in some years that it is difficult to imagine that, without effective testing, many countries have remained free of pospivioid infections.

Before 2012, when testing tomato seed imports to Australia became mandatory, it was only anticipated that PSTVd-infected seed lots might be detected. Detections of the other pospiviroid species were unexpected. Testing methods that were capable of detecting the wider set of species were used. This experience supports the view that broad-specificity detection methods that are also sensitive, such as next generation sequencing, should be considered for quarantine purposes as they are more likely to detect unexpected species.

Infected seed lots detected through the Australian testing were either destroyed or re-exported. The seed supply probably changed in response to these adverse detections and regulatory actions, which may explain why there were changes in the frequency of detections.

The size of the samples taken for testing allowed contamination at very low levels to be detected in some lots. The very low levels of contamination needs to be explained and considered in industry practices. These levels make detection difficult, and when infected seeds escape detection they are likely to influence epidemiology. This perspective validates the quarantine measure imposed by the Australian government requiring the testing of primary samples of up to 20,000 seeds. If smaller samples had been tested by the Australian laboratories, the small numbers of infected seeds would not have been detected, but the threat of infection and invasion by pospiviroids would have remained.

## Figures and Tables

**Figure 1 viruses-11-00098-f001:**
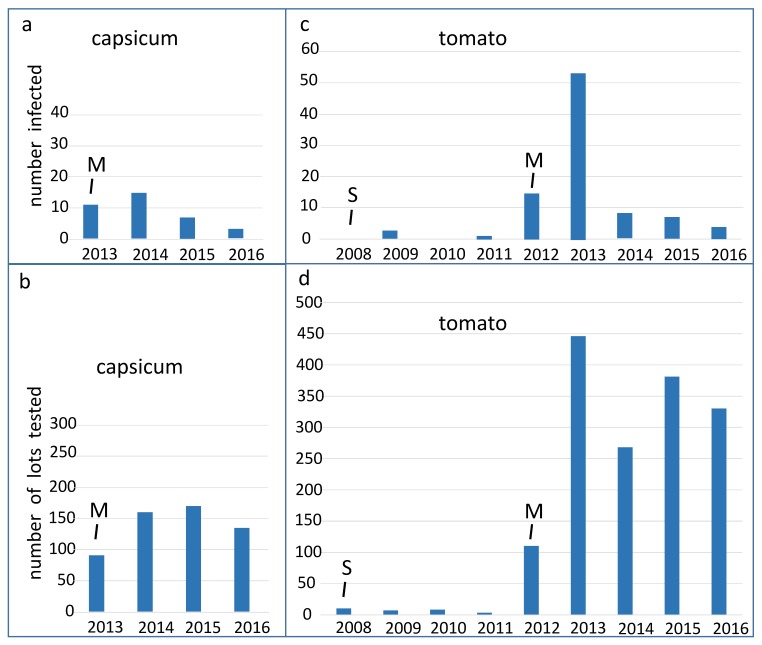
Variation in the numbers of infected seed lots detected each year. (**a**) The numbers of infected capsicum seed lots and (**b**) the numbers of capsicum seed lots tested each year by Australian laboratories, and (**c**,**d**) equivalent counts of infected tomato seed lots and numbers of tomato seed lots tested. Testing of tomato seed for *Potato spindle tuber viroid* (PSTVd) started in 2008 (S) but only a small fraction of imports was tested. Testing of tomato seed imports became mandatory in 2012 (M), and the full suite of tests for the viroids was introduced early in 2013, but data was not compiled for all tested seed lots until 2013. Mandatory testing of capsicum (chilli, pepper and capsicum) seed was introduced in 2013 (M).

**Figure 2 viruses-11-00098-f002:**
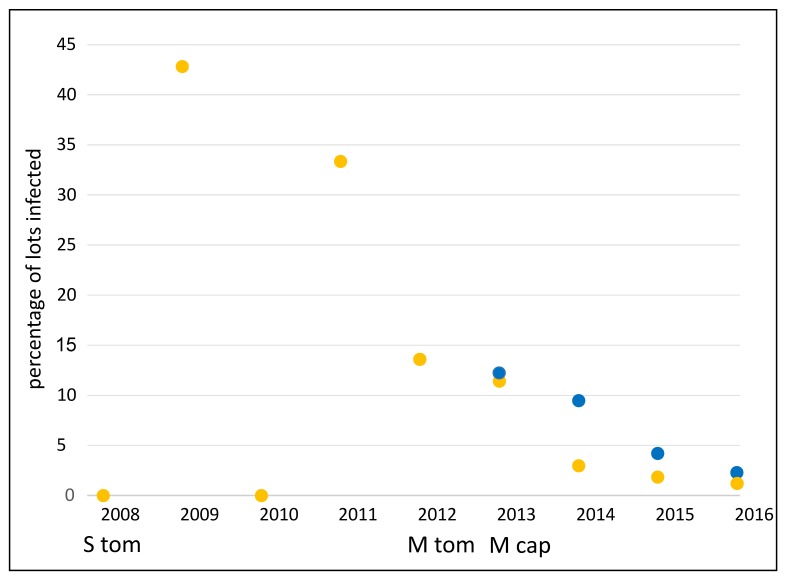
Proportions of tested capsicum and tomato lots (blue and yellow symbols respectively) found to be infected each year. Testing of tomato seed for PSTVd started in 2008 (S tom). Testing of tomato seed imports became mandatory in 2012 (M tom). Testing of capsicum seed was introduced in 2013 and was mandatory from the start (M cap).

**Figure 3 viruses-11-00098-f003:**
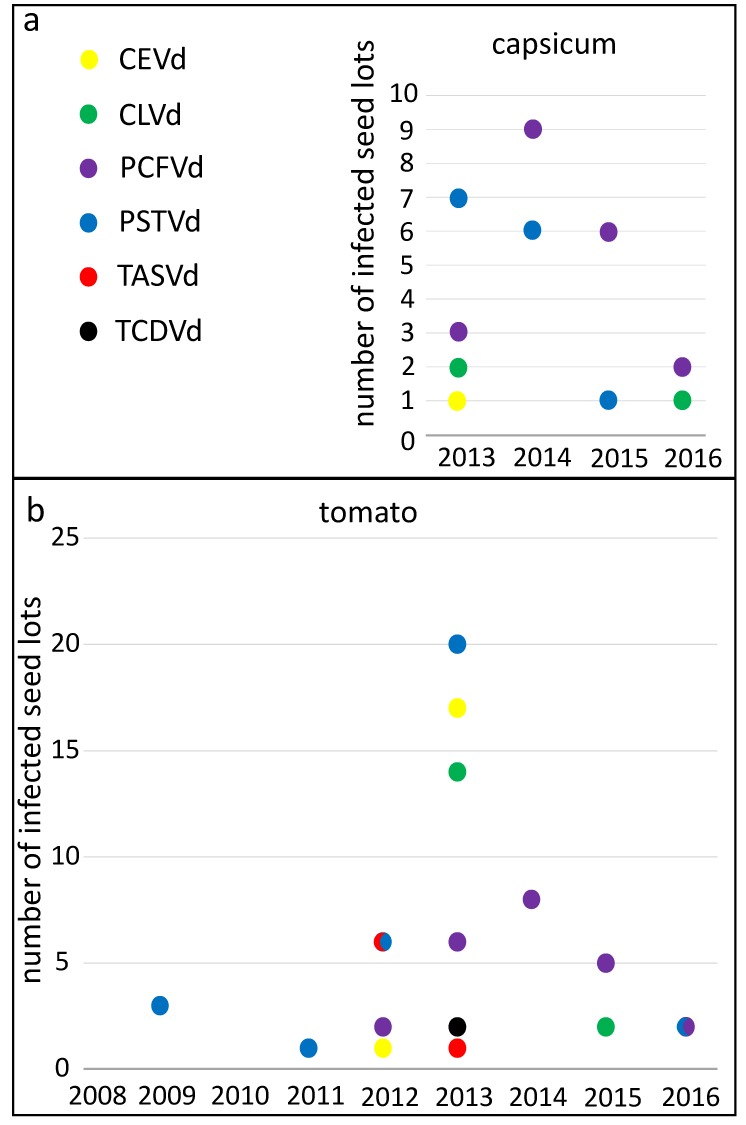
Detections of viroid species in capsicum and tomato seed lots by year. Numbers of detections of viroid species in capsicum (**a**) and tomato (**b**) seed lots. PSTVd (blue) and *Tomato apical stunt viroid* (TASVd) (red) were both detected six times in tomato seed lots in 2012, and *Pepper chat fruit viroid* (PCFVd) (purple) and PSTVd (blue) were both detected twice in tomato seed lots in 2016.

**Figure 4 viruses-11-00098-f004:**
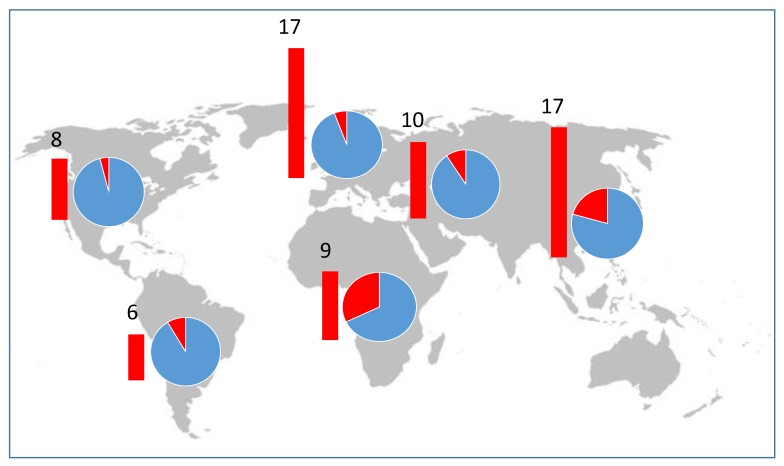
Exports of infected seed lots from major seed production regions. Numbers of infected seed lots (red bars) and the proportions of infected seed lots (pie charts), as calculated from the number of infected lots and the total known number of lots tested. Countries were identified from certification documentation that arrived with the seed lots. Seed lots were only included in this dataset when the production country data was collected.

**Figure 5 viruses-11-00098-f005:**
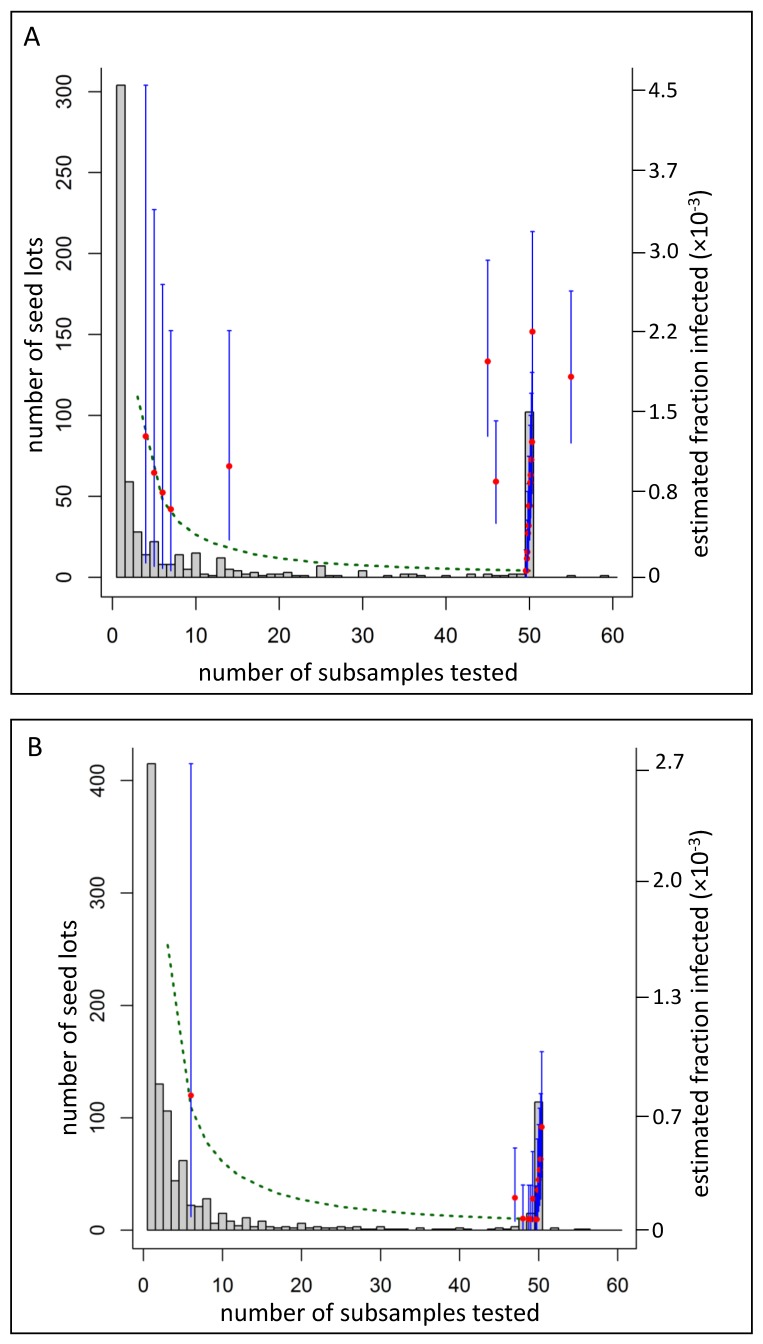
Estimates of the fraction of infected seeds (red points) in PSTVd-infected seed lots detected by the EMAI (**a**) and CHS (**b**) laboratories. The size of the seed lots (*x*-axis) is represented by the numbers of subsamples tested from the lot. Credible intervals (blue bars) and the minimum possible best estimates given the number of subsamples tested (green dotted line) are shown. Seed lots (*x*-axis) were bi-modally distributed in terms of sizes (grey bars): many lots were comprised of either fewer than 4000 seeds (<=2 subsamples) or 100,000 seeds or more (50 or more subsamples).

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
