# Peer review of "Viroid-infected Tomato and Capsicum Seed Shipments to Australia"

_viruses, 2019, doi:10.3390/v11020098_

Reviewer 1 Report

The manuscript titled "Viroid-infected tomto and capsicum seed shipments to Australia" presentes an analysis of the results of several years worth of statutory testing for the presence of solanaceous pospiviroids in seed lots being traded into Australia. The manuscript is generally well written, data are clearly presented, and the discussion and conclusions are concisely argued without the key messages getting lost.

Although these data are not the results of a stratified survey, and consequently present a 'self-selecting' dataset, trade related data such as these are invaluable in gaining a deeper understanding of  the epidemiology of these pathogens. The authors should be commended for crrying out this type of analysis.

There are some areas where the text needs minor review, though this is largely semantics. There are also areass where the manuscript may benefit from expanding elements of discussion points.

This publication will be of interest both to virologists and plant health policy makers. In light of this a suggestion would be that the text should be reviewed to bring terminology in line with ISPM 05, the glossary on phytosanitary terms.  Most noticable is the use of the word 'Invasion'. The term 'incursion' would be preferable. Incursion is used later in the text, but this should be used more consistently.

Lines 18-20 (Abstract) - redraft  to include specific years testing has been carried out for accuracy.

Line 61 : Insert the word 'phytosanitary' to read 'and the phytsanitary preparation of seed lots'

Line 120: Change 'was' to 'were'

Line 133-136 : Is there any evidence to suggest that seed companies have improved their own test methods? This could help explain a drop in interceptions.

Line 160-168: The authors explain the lower averge rate of detection in the Bruinsma study as a possible result of different detection method or in seed sources, or in sample size. Given the results of this study this would more strongly indicate that sample size has the greatest bearing. Especially given the relative increase in sensitivity reported for testing by real-time RT PCR (TaqMan). Please redraft for clarity.

As an aside and not necessarily for inclusion in this manuscript: this supposition could be tested from data generated in this study through analysing selections of random subsamples from positive seed lots to approximate testing of smaller seed lot sample sizes used in previous studies. 

Line 189: Detection of a broader range of pospiviroids occurred partly (?) because of using broader range primer sets. What other contributing factors were there?

Line 227-247: Geographic origin of positive findings presented is largely presented from the perspective of origin of seed. There is a suggestion that phylogenetic analysis was carried out. It would be of interest to see whether the sequence of viroid detected appears to corroborate the origin of seed, however, it is appreciated that in some areas such as Africa the scant availabilty of reference sequence may limit this type of analysis.

Line 282-285: Is there opportunity here to also discuss the implictions from this work to the sample sizes recommended by international standard setting bodies such as ISF-ISHI-Veg and ISTA. Whilst pospiviroids are not currently covered in these standards, they do cover other seed transmitted viral pathogens and it is envisaged that these standards could expand to cover viroids.

Line 326-327: Add references to support the opening statement of the conclusions (10,11?)

Line 500-511: References 50-55 look too be standard formatting examples and should be removed.

Author Response

We (the authors) appreciated the work of the reviewers and would like to thank them for their help. Our responses are given in blue type and are marked with two dashes '--'.  The rest of the text (in black) is copied from the review.

There are some areas where the text needs minor review, though this is largely semantics. There are also areass where the manuscript may benefit from expanding elements of discussion points.

--A senior scientific editor who works for the Australian government has looked over the paper and suggested several edits of the English. Edits have been made and tracked on the revised version. Discussion points have been expanded consistent with the other requests made by the reviewer (see below).

This publication will be of interest both to virologists and plant health policy makers. In light of this a suggestion would be that the text should be reviewed to bring terminology in line with ISPM 05, the glossary on phytosanitary terms.  Most noticable is the use of the word 'Invasion'. The term 'incursion' would be preferable. Incursion is used later in the text, but this should be used more consistently.

--No change has been made. We looked at every instance where the words ‘invasion’ or ‘incursion’ were used and concluded that they were used appropriately in every case. The word ‘invasion’ is used in a manner that is consistent with the literature on invasive species (eg Sax & Gaines PNAS August 12, 2008 105  11490 -11497) and the Convention on Biological Diversity - COP 6 Decision VI/23A. The term ‘invasion’ is commonly taken to include instances when an exotic species becomes permanently established in an area (naturalised) as well as instances where an exotic species is introduced and a transient population exists for a limited time. The International Plant Protection Convention defines an incursion more narrowly to be ‘An isolated population of a pest recently detected in an area, not known to be established, but expected to survive for the immediate future’ (ISPM 5).

Lines 18-20 (Abstract) - redraft  to include specific years testing has been carried out for accuracy.

--A correction has been made in accordance with the reviewer’s request (Lines 18-20).

Line 61 : Insert the word 'phytosanitary' to read 'and the phytsanitary preparation of seed lots'

--A correction has been made in accordance with the reviewer’s request (Line 63).

Line 120: Change 'was' to 'were'

--A correction has been made in accordance with the reviewer’s request. The singular or the plural could be appropriate in this sentence. We suggest the editor look at it. (Line 121).

Line 133-136 : Is there any evidence to suggest that seed companies have improved their own test methods? This could help explain a drop in interceptions.

--Additional information has been added in response to the reviewer’s suggestion (Lines 138-142). We do not know if the seed companies improved their testing methods, but two major seed testing laboratories did change their methods, presumably improving them.

Line 160-168: The authors explain the lower averge rate of detection in the Bruinsma study as a possible result of different detection method or in seed sources, or in sample size. Given the results of this study this would more strongly indicate that sample size has the greatest bearing. Especially given the relative increase in sensitivity reported for testing by real-time RT PCR (TaqMan). Please redraft for clarity. 

--Minor changes have been made in accordance with the reviewer’s request (Line 168 & 173).

As an aside and not necessarily for inclusion in this manuscript: this supposition could be tested from data generated in this study through analysing selections of random subsamples from positive seed lots to approximate testing of smaller seed lot sample sizes used in previous studies.  

--We agree with the reviewer that this kind of experiment could be done, but the mathematics of the different sample sizes is fairly straight forward and uncontroversial, and so the experiment would only be confirmatory.

Line 189: Detection of a broader range of pospiviroids occurred partly (?) because of using broader range primer sets. What other contributing factors were there?

--Additional text has been added consistent with the reviewer’s suggestion (Line 207). The larger primary sample is probably also a factor.

Line 227-247: Geographic origin of positive findings presented is largely presented from the perspective of origin of seed. There is a suggestion that phylogenetic analysis was carried out. It would be of interest to see whether the sequence of viroid detected appears to corroborate the origin of seed, however, it is appreciated that in some areas such as Africa the scant availabilty of reference sequence may limit this type of analysis.

--No change has been made. The reviewer is correct that our analysis of geographic data is almost entirely based on the reported origins of the seed. At this time, it isn’t clear to us that phylogenetic analyses will produce much more information about the geographic distributions of the viroids. We have done some phylogenetics, but to complete that work will require considerable time and staff resources that we do not have. Viroids are highly-structured, very-short RNAs, so unusual alignment parameters are required and the sequences carry little phylogenetic signal. We have found that often sequences from the same species cannot be distinguished phylogenetically, as there is typically poor support for the internal branches within a species cluster.

Line 282-285: Is there opportunity here to also discuss the implictions from this work to the sample sizes recommended by international standard setting bodies such as ISF-ISHI-Veg and ISTA. Whilst pospiviroids are not currently covered in these standards, they do cover other seed transmitted viral pathogens and it is envisaged that these standards could expand to cover viroids.

--Additional text and another reference have been added consistent with the reviewer’s suggestion (Lines 304-305; reference 49). Reference 49 is a method for testing for pospiviroids in tomato seed published by the International Seed Federation. The ISF method argues for a sample of only 3000 seeds. We discuss this method and the small sample size it recommends in the new text we have added.

Line 326-327: Add references to support the opening statement of the conclusions (10,11?) 

--References have been added (Line 350, references 10 and 49).

Line 500-511: References 50-55 look too be standard formatting examples and should be removed.

--The reference lines have been removed in accordance with the reviewer’s request (Lines 533-544).

Reviewer 2 Report

This manuscript describes the very important issue of virus/viroid quarantines in the world. Their results are clear-cut and the authors have spent a long time of viroid detection in the chili/pepper seeds and other solanaceous seeds. it is always a debate issue that the new or infected/detected viruses are derived from seed contamination or natural infection via vectors or mechanical wounds. In fact, this manuscript will offer a good case for readers since there are few papers for seed quarantine. 

Author Response

We (the authors) greatly appreciated the work of the reviewers and would like to thank them for their help. We have not responded to any particular points raised by the reviewer.

Reviewer 3 Report

The manuscript discusses the spread of Pospiviroid species through infected tomato and capsicum seeds to new geographic regions, which may lead to a potential disease epidemic and pose a growing threat to local crops.

The authors present the first report of large-scale screening of commercially traded capsicum seeds for pospiviroids and a more detailed report regarding their detection rate in tomato seeds. The study shows the presence of 6 representatives of Pospiviroids in commercial tomato seed lots, four of which were detected also in capsicum seed lots. Sequencing analysis of positive signals in viroid detection found variants from each viroid species. New geographic distribution to Middle East was reported for CLVd.    

The results showed that the low titer of viroids in infected seeds requires a screening of a large set of seeds and application of more sensitive methods of detection as next-generation sequencing.  

Major revision:

The authors have to provide the data of variant sequences and comparison analysis. Related to this the authors claim that sequence variants have been detected but they should provide a proof that the observed variants are not due to PCR infidelity.  

The phylogenetic analysis of PSTVd isolates should be presented as a supplementary data.

Minor remarks:

Please, avoid redundant information on fig.1 and fig.2 by keeping only fig.2.

Line 194 According to the figure 3, the PSTVd’ peak year for tomato is 2013 not 2012.

Author Response

We greatly appreciate the work of the reviewers and would like to thank them for their help.  Our responses are given in blue type and marked with two dashes ('--'). The rest of the text (in black) is copied from the review.

Major revision:

The authors have to provide the data of variant sequences and comparison analysis. Related to this the authors claim that sequence variants have been detected but they should provide a proof that the observed variants are not due to PCR infidelity.  

--We have included a short paragraph that accounts for the PCR misincorporation problem raised by the reviewer (Lines 190-200), and we are now submitting supplementary files that provide information that we hope will satisfy the reviewer. Supplementary file 3 provides a fair number of our sequences, and supplementary files 1 and 2 report examples of some of our sequence analyses. We could only attach one of the supplementary files with this response.

--We think that three main findings reported in these supplementary files (1 and 2) should resolve this question about variants and PCR infidelity. First, we obtained sequences that were identical to sequences of isolates in the GenBank database, or that closely matched sequences in GenBank. Second, we obtained sequences from different seed lots that were identical or nearly identical. We describe both kind of matches in the supplementary files very briefly and provide some alignments. Third, we obtained sequences from different seed lots that matched different clusters of variants (isolates) in the database. Taken together, we think these three general findings indicate that our RT-PCR amplifications and sequencing generated reasonably accurate sequence that was adequate to distinguish variants.

--We agree with the reviewer that variation due to PCR misincorporation must be considered. The literature indicates error rates from PCR misincorporation can be as high as 5% (in the worst cases), but usually the rate is much lower (Eckert and Kunkel, Genome Res. 1991 1: 17-24;  BrachoMoyaBarrio 1998 Journal of General Virology 79: 2921-2928). In addition to our view of the sequence matches, we think the sequences are likely to be fairly accurate because they were relatively short (no longer than 369 nts) and were obtained in both directions from each amplified product, and primer sequences and terminal stretches, that were poor quality, were removed, as per normal practice.

The phylogenetic analysis of PSTVd isolates should be presented as a supplementary data.

--We do not believe that phylogenetic work is required for this paper. We have removed the mention of a phylogenetic analysis from the text (Lines 263-265), as the phylogeny mentioned in the previous version was not found using ideal methods. Phylogenetic analysis of viroids poses special problems as they are highly-structured, very-short RNAs. Unusual alignment parameters are required and the sequences carry little phylogenetic signal. We have done some phylogenetic analyses and found that often sequences from the same species cannot be distinguished phylogenetically, as there is typically poor support for the internal branches within a species cluster.

Minor remarks:

Please, avoid redundant information on fig.1 and fig.2 by keeping only fig.2.

--No change was made. We believe Figure 1 should be kept in the paper as it provides a helpful picture of the numbers of seed lots tested, numbers of detections, yearly variations in the numbers, and differences between the tests on capsicum and tomato. We think the readers will appreciate having this picture as it will help their understanding of the trends we discuss and how they relate to the regulatory changes. Figure 2 is very different as it provides information about the proportion of infected seed lots in a way that is quite abstract.  We believe Figure 2 is most easily understood when Figure 1 can also be examined.

Line 194 According to the figure 3, the PSTVd’ peak year for tomato is 2013 not 2012.

--A correction has been made in accordance with the reviewer’s request (Line 210).

Round  2

Reviewer 1 Report

The authors have made significant improvements in this manuscript from the previous submission.I congratulate them on a significant body of work.

Reviewer 3 Report

I consider that the last version of the manuscript can be accepted for publication.